# Co-Amorphization, Dissolution, and Stability of Quench-Cooled Drug–Drug Coamorphous Supersaturating Delivery Systems with RT-Unstable Amorphous Components

**DOI:** 10.3390/pharmaceutics16121488

**Published:** 2024-11-21

**Authors:** Yan-Fei Zhang, Qian Yao, Xiao-Ying Lin, Ying-Hui Ma, Hui-Feng Zhang, Huan Yu, Shang-Qiang Mu, Chuang Zhang, Hao Geng, Cheng-Yi Hao, Li-Li Zuo, Di Wu, Yue Li, Li-Li Jin, Nian-Qiu Shi

**Affiliations:** 1School of Pharmacy, Jilin Medical University, Jilin 132013, China; zyf20181812@163.com (Y.-F.Z.);; 2College of Pharmaceutical Sciences, Yanbian University, Yanji 133002, China; 17798214832@163.com (Q.Y.);; 3Affilittend Hospital of Jilin Medical University, Jilin 132011, China

**Keywords:** supersaturating drug delivery systems (SDDS), “drug–drug” coamorphous system (ddCAM), quench cooling, coamorphization and stability, combination therapy

## Abstract

**Background**: Supersaturating drug delivery systems (SDDSs) have gained significant attention as a promising strategy to enhance the solubility and bioabsorption of Biopharmaceutics Classification System (BCS) II drugs. To overcome challenges associated with polymer-based amorphous SDDS (aSDDS), coamorphous (CAM) systems have emerged as a viable alternative. Among them, “drug-drug” CAM (ddCAM) systems show considerable potential for combination drug therapy. However, many drugs in their pure amorphous forms are unstable at room temperature (RT), complicating their formation and long-term stability profiles. Consequently, limited knowledge exists regarding the behavior of ddCAMs containing RT-unstable components formed via quench cooling. **Methods**: In this study, we used naproxen (NAP), a RT-unstable amorphous drug, in combination with felodipine (FEL) or nitrendipine (NTP), two RT-stable amorphous drugs, to create “FEL-NAP” and “NTP-NAP” ddCAM pairs via quench cooling. Our work used a series of methods to perform a detailed analysis on the co-amorphization, dissolution, solubility, and stability profiles of ddCAMs containing RT-unstable drugs, contributing to advancements in co-amorphization techniques for generating SDDS. **Results**: This study revealed that the co-amorphization and stability profiles of ddCAMs containing RT-unstable components produced via a quench-cooling method were closely related to drug-drug pairing types and ratios. Both quench-cooling and incorporation into coamorphous systems improved the dissolution, solubility, and physical stability of individual APIs. **Conclusions**: Our findings provide deeper insight into the co-amorphization, dissolution, and stability characteristics of specific drug-drug coamorphous systems FEL-NAP and NTP-NAP, offering valuable guidance for developing new ddCAM coamorphous formulations containing some RT-unstable drugs.

## 1. Introduction

Poor water solubility is a major challenge in the development of many active pharmaceutical ingredients (APIs) and newly discovered drug candidates, as it negatively impacts bioavailability and bioabsorption in vivo [1,2,3]. Most active pharmaceutical ingredients (APIs) fall under Biopharmaceutics Classification System (BCS) classes II and IV, with class II drugs receiving particular attention due to their low solubility, which poses a significant challenge after oral administration, despite their relatively good intestinal permeability [4,5].

Supersaturating drug delivery systems (SDDS) have emerged as a promising approach for enhancing the bioavailability of poorly soluble APIs [6,7]. By inducing a supersaturated state, SDDS maintain elevated solubility levels in the gastrointestinal tract for extended periods, which facilitates improved absorption, particularly for BCS class II and IV drugs. Amorphization, the conversion of a drug from its crystalline to its amorphous form, is an effective strategy for enhancing aqueous solubility [8,9]. Amorphous solid dispersions (ASDs), a type of amorphous SDDS (aSDDS), are widely used to formulate poorly water-soluble candidates [10,11].

Nevertheless, rapid crystallization or precipitation often occurs when amorphous drugs are stored in the absence of crystallization inhibitors, due to fast nucleation and crystal growth. Polymers are commonly used as carriers or precipitation inhibitors, suppressing crystal growth and forming solid glass solutions through molecular dispersion. Some amorphous polymers, such as Soluplus^®^ and hydroxypropyl methylcellulose (HPMC), have demonstrated beneficial solubilization properties and the ability to maintain the supersaturated state [12,13]. However, polymer hygroscopicity can lead to moisture-induced crystallization, compromising the stability of the amorphous form. Additionally, excessive polymer use may result in large dosage volumes, which can limit practical applications [14].

Coamorphous (CAM) systems have been developed to address these challenges, owing to their advantages over traditional supersaturable ASDs [15]. There are two kinds of CAMs: “drug-excipient” CAMs (deCAMs), which use small molecular co-formers such as amino acids, organic acids, or surfactants [15], and drug–drug CAMs (ddCAMs), which employ combinations of two pharmacologically relevant drugs [16]. ddCAMs, in particular, provide potential benefits for combination therapies.

The fabrication of amorphous SDDS commonly involves two main disordering processes: thermodynamic and mechanical. Thermodynamic disordering utilizes a thermodynamically stable, non-crystalline form of a drug (e.g., through melting and rapid cooling) to induce amorphization [17]. In contrast, mechanical disordering transforms crystalline drugs into amorphous forms via techniques like ball-milling or cryo-milling, which are “top-down” approaches [18]. While both methods can be used to fabricate CAMs, amorphous SDDSs produced through mechanical processes tend to be less physically stable than those formed through thermodynamic methods. Moreover, mechanical processes often result in incomplete molecular mixing, leading to relaxation, nuclei formation, and crystal growth. Thermodynamic processes, on the other hand, achieve molecular-level mixing, reducing the risk of crystallization. Quench-cooling-based thermodynamic processes, unlike solvent evaporation, avoid solvents and the associated risk of solvent-induced phase transformations. This approach is especially suited for thermostable drugs, as it allows single-phase molecular mixing in the molten state during fabrication. Some ddCAM formulations produced using quench cooling have shown enhanced physical stability [14,15].

Quench cooling is a widely used method for preparing glassy formulations in ddCAM formulations, offering improved stability. However, some drugs remain unstable in their amorphous form at room temperature (RT), limiting the formation and stability of ddCAMs [19]. This challenge has hindered the development of ddCAMs containing RT-unstable drugs.

In this study, we utilized naproxen (NAP), an RT-unstable amorphous drug with a glass transition temperature (*T*_g_) of 278.19 K, in combination with felodipine (FEL) and nitrendipine (NTP), which are RT-stable drugs, to form ddCAM pairs [19]. The chemical structures are shown in Figure 1A–C. The drug combinations “FEL-NAP” and “NTP-NAP” were developed as potential combination therapies by a quench-cooling-based thermodynamic disordering process. These drug pairs in this study hold therapeutic potential for managing both hypertension and pain [20,21]. Additionally, both of these BCS class II drugs benefit from enhanced solubility through dual amorphization, facilitating their combined delivery [4]. Key formation characteristics of these ddCAMs were evaluated using PXRD, DSC, and SEM to assess amorphous states and micromorphology. We also investigated the dissolution, solubility, stability, and intramolecular interaction of the ddCAM pairs. This research provides a comprehensive understanding of co-amorphization, dissolution, solubility, and stability profiles for some specific ddCAM systems.

## 2. Materials and Methods

### 2.1. Materials

FEL, NTP, and NAP were obtained from Kangbaotai Fine Chemicals Co., Ltd. (Wuhan, China), with purities exceeding 99%. Methanol was purchased from Tianjin Yuwang Reagent Co. (Tianjin, China). All other chemicals and reagents were of analytical grade. All drugs and other materials were used without modification.

### 2.2. Preparation of ddCAMs via Quench Cooling

To prepare coamorphous systems, drug–drug pairs of FEL and NAP, as well as NTP and NAP, were mixed thoroughly for 10 min in plastic bags at weight ratios of 1:1, 1:2, and 2:1 to create physical mixtures. Known amounts of these mixtures were transferred to porcelain dishes and heated to generate homogeneous fused liquids by oil bath. The fused liquids were then rapidly quench-cooled in liquid nitrogen at −196 °C to produce crude quench-cooled systems. Pure amorphous forms of the individual drugs were also prepared using the same quench-cooling-based method. The crude systems were ground using a mortar and pestle, and clumps were removed by passing the material through an 80-mesh sieve. Powdered quench-cooled systems were stored in a desiccator until needed for experiments. Abbreviations for all samples are provided in Table 1.

### 2.3. PXRD Analysis

PXRD analysis was performed to evaluate the quench-cooled systems and control samples using a D/Max-2500 X-ray fluorescence spectrometer (Rigaku, Tokyo, Japan). The CuKα radiation source (λ = 1.541 Å) was operated at 40 kV and 100 mA. Scans were conducted at a rate of 10°/min over a 2*θ* range of 5°–45°, with a step size of 0.04° 2*θ* and a counting time of 0.5 s/step.

### 2.4. SEM Analysis

The surface morphologies of quench-cooled systems and control samples were observed via SEM (JSM-6490LV, JEOL, Tokyo, Japan). SEM images were captured at an operating voltage of 20 kV. Samples were mounted onto glass stubs with double-sided adhesive tape. The mounted samples were coated with gold under vacuum in an argon atmosphere before imaging. The resulting micrographs provided information on the morphological and surface features of the samples.

### 2.5. DSC Analysis

The amorphous nature of the prepared samples was assessed using a DSC system (SDT-Q600, TA Instruments, Milford, MA, USA). The instrument was calibrated with indium as a standard. The samples include crystalline FEL, crystalline NAP, FEL-NAP_PM_ (1:1), amorphous FEL, FEL-NAP (1:1), FEL-NAP (1:2), FEL-NAP (2:1), crystalline NTP, crystalline NAP, NTP-NAP_PM_ (1:1), amorphous NTP, NTP-NAP (1:1), NTP-NAP (1:2), and NTP-NAP (2:1). Approximately 5 mg of each sample was placed into aluminum pans, and then DSC analysis was performed under a nitrogen purge at a scan rate of 10 °C/min over a temperature range of 20–250 °C.

### 2.6. FT-IR Spectroscopy

An FT-IR spectroscopy system (Nicolet IS5, Thermo Scientific, Waltham, MA, USA) was used to evaluate intermolecular interactions between FEL, NTP, and NAP within the coamorphous systems. Each sample (~2–3 mg) was mixed with dry potassium bromide and pressed into a pellet using a compression tool (FW-4A, Tianjin, China). Spectra were recorded in the range of 4000 cm^−1^ to 400 cm^−1^ with a resolution of 1 cm^−1^.

### 2.7. HPLC Method for Drug Qualification

The chemical stabilities of FEL, NTP, NAP, and all quench-cooled systems were analyzed using a Shimadzu high-performance liquid chromatography (HPLC) system consisting of an LC-20AT pump, a UV–VIS tunable absorbance detector, an autosampler, an in-line degasser, and a reverse-phase C18 column (250 × 4.6 mm, C18). The mobile phases, detection wavelength, and calibration range are separately shown in Table 2. The flow rate was 1.0 mL/min. Samples were filtered using 0.45 μm membrane filters before analysis. HPLC was performed using a 20 μL injection volume, and each run lasted 10 min.

### 2.8. Stability Assay of the Amorphous State

Amorphous FEL, amorphous NTP, and all coamorphous systems were maintained under constant temperature (25 °C) and relative humidity (45 ± 5%) conditions. PXRD profiles were measured at 0 and 1 month of storage using the aforementioned PXRD method (Section 2.3).

### 2.9. Dissolution Testing

To evaluate drug release profiles for quench-cooled systems and controls, dissolution testing was conducted at 37 °C using a ZRS-8G dissolution apparatus (Tianda Tianfa-pharmaceutical testing instrument manufacturer is located in Tianjin, China) with a paddle rotation speed of 100 rpm. The FEL-related samples, NTP-related samples, and NAP-related samples with respective mass equivalents (50 mg) of FEL, NTP, and NAP were added to 900 mL of an aqueous 0.2% SDS solution. The samples were then directly placed in the dissolution apparatus, and aliquots were taken at specified time points (5, 10, 15, 30, 45, and 60 min), filtered through 0.45 μm Millipore filters, and protected from light to prevent photodegradation. Each filtrate was diluted and analyzed by HPLC-UV as described in Section 2.7. Diluted filtrates were immediately tested for FEL, NTP, and NAP content since recrystallization was a concern due to the lower temperature outside the apparatus.

### 2.10. Solubility Testing

Saturation solubilities of quench-cooled systems and controls were assessed by shaking excess sample (~50 mg) in 50 mL of water at 37 °C for 24 and 48 h in a thermostatically controlled gas bath shaker. Samples were filtered through 0.45 μm Millipore filters, diluted, and analyzed by HPLC as described in Section 2.7.

### 2.11. Data Analysis

Statistical analysis was performed using an unpaired Student’s *t*-test, with results expressed as the mean ± standard deviation. Statistical significance was indicated by *p* < 0.001, *p* < 0.01, and *p* < 0.05.

## 3. Results and Discussion

### 3.1. HPLC Analysis

Co-amorphous ddCAMs have gained considerable attention for their ability to improve the dissolution behavior of BCS II drugs. While notable progress has been made in enhancing the physical stability and solubility of drugs through ddCAMs, further investigation is required, particularly for ddCAM pairs containing RT-unstable amorphous drugs. Once the co-amorphization process is complete, chemical stability becomes a critical factor, ensuring that the drugs retain their chemical structures and activities during storage and use, thereby preventing degradation. Thus, evaluating the chemical stability of ddCAMs is essential to their successful development and application.

The HPLC chromatograms of monomer FEL, NAP, NTP, and their mixture are shown in Figure 2. As shown in Table 3, the FEL and NAP content in the FEL-NAP systems prepared at ratios of 1:1, 1:2, and 2:1 were consistent with the crystalline forms of both drugs. The FEL content was more than 98% for the 1:1, 1:2, and 2:1 FEL-NAP systems. Correspondingly, the NAP content was also above 98%. Similarly, in the NTP-NAP systems, the NTP and NAP content at the same ratios aligned with their crystalline counterparts (Table 4). The NTP content was maintained above 97%, while the NAP content was 96%. Although the NAP content in the NTP-NAP systems exhibited a slight decrease, it remained above 95% in all cases. Overall, these results demonstrate the high uniformity of drug content across all quench-cooled samples, providing a solid foundation for further DSC, PXRD, FT-IR, SEM, dissolution, and solubility analyses.

### 3.2. PXRD Analysis Result

Figure 3A presents the diffractograms for crystalline FEL, crystalline NAP, FEL-NAP_PM_, FEL-NAP (1:1), FEL-NAP (1:2), and FEL-NAP (2:1). The X-ray diffractogram of crystalline FEL exhibited sharp peaks at diffraction angles (2*θ*) of 10.44°, 16.4°, 22.0°, 23.48°, 25.56°, 26.6°, and 32.84°, indicating a typical crystalline structure. Similarly, the diffractogram for crystalline NAP showed distinct peaks at 2*θ* angles of 6.76°, 12.72°, 13.48°, 16.88°, 18.08°, 19.16°, 20.44°, 22.56°, 23.88°, 27.52°, 28.6°, and 31.48°, confirming its crystalline form. As expected, amorphous FEL displayed no discernible diffraction peaks, indicating its non-crystalline structure. FEL-NAP_PM_ exhibited diffraction peaks at 2*θ* angles 6.88°, 10.48°, 12.92°, 16.48°, 19.24°, 20.68°, 22.6°, 23.56°, 25.56°, 26.8°, and 32.24°, similar to those of FEL and NAP. Notably, no diffraction peaks were observed for the coamorphous systems FEL-NAP (1:1), FEL-NAP (1:2), and FEL-NAP (2:1), suggesting successful formation of amorphous phases through the quench-cooling process, thereby confirming the formation of ddCAM systems.

Figure 3B shows the diffraction patterns for crystalline NTP; crystalline NAP; NTP-NAP_PM_; and the quench-cooled systems NTP-NAP (1:1), NTP-NAP (1:2), and NTP-NAP (2:1). The diffractogram of crystalline NTP exhibited sharp peaks at 2*θ* angles of 8.32°, 10.56°, 11.96°, 14.88°, 16.44°, 17.36°, 18.56°, 19.96°, 22.68°, 24.88°, 26.12°, 27.16°, and 28.08°, confirming its crystalline structure. Amorphous NTP, by contrast, showed no clear diffraction peaks, consistent with its amorphous state. The NTP-NAP_PM_ diffractogram displayed peaks similar to those of NTP and NAP, with diffraction angles at 6.88°, 10.2°, 11.52°, 13.24°, 16.92°, 19.16°, 22.56°, 23.92°, 24.52°, 26.12°, 27.64°, and 28.72°. Interestingly, while the quench-cooled systems NTP-NAP (1:1) and NTP-NAP (1:2) exhibited diffraction patterns resembling those of crystalline NTP and NAP, no diffraction peaks were observed for NTP-NAP (2:1), confirming its completely amorphous form and ddCAM formation.

PXRD analysis of the fabricated coamorphous systems showed no crystalline peaks, further verifying their amorphous nature, as supported by DSC data. These findings demonstrate that FEL-NAP (1:1), FEL-NAP (1:2), FEL-NAP (2:1), and NTP-NAP (2:1) successfully formed coamorphous ddCAM systems, suggesting that the formation of coamorphous “drug–drug” systems may depend on the specific ratios and pairings of drug components.

### 3.3. Micromorphology

The micromorphologies of crystalline FEL, NTP, and NAP powders, along with FEL-NAP_PM_, NTP-NAP_PM_, FEL-NAP (1:1), and NTP-NAP (2:1), are shown in Figure 4A. Crystalline FEL exhibited distinctive strip-block structures, clearly visible at ×100 magnification (Figure 4A), while NTP displayed similar block-like structures (Figure 4B). In contrast, NAP particles possessed a spheroidal microstructure (Figure 4C). Both strip-block and spheroidal microstructures were observed in the FEL-NAP_PM_ (Figure 4D) and NTP-NAP_PM_ (Figure 4F), while the micromorphologies of FEL-NAP (1:1) and NTP-NAP (2:1) systems, prepared using the same manufacturing method, appeared as irregular block shapes (Figure 4E,G). Additionally, the crystalline FEL, NTP, and NAP powders, as well as the FEL-NAP_PM_ and NTP-NAP_PM_ formulations, exhibited irregular micromorphologies. These observations offer valuable insights into the physical characteristics of quench-cooled systems.

### 3.4. Melting Point and T_g_ Changes

The DSC thermographs of the quench-cooled systems are presented in Figure 5 (or Appendix A), covering a temperature range from 50 °C to 240 °C. For crystalline FEL, NAP, and FEL-NAP_PM_ (Figure 5A), sharp endothermic peaks were observed at 149.11 °C, 161.35 °C, and 132.07 °C, respectively, corresponding to their melting points. In contrast, no melting peaks were detected for amorphous FEL or the FEL-NAP (1:1), FEL-NAP (1:2), and FEL-NAP (2:1) systems. Similarly, for crystalline NTP, NAP, and NTP-NAP_PM_ (Figure 5B), sharp endothermic peaks were noted at 176.21 °C, 161.35 °C, and 142.41 °C, respectively. However, no melting peaks were observed for amorphous NTP or the NTP-NAP (2:1) system. These DSC results confirm that FEL-NAP (1:1), FEL-NAP (1:2), FEL-NAP (2:1), and NTP-NAP (2:1) systems are completely amorphous, as indicated by the absence of endothermic peaks, which suggests the formation of a homogeneous phase.

The experimental *T*_g_ values of the FEL-NAP (1:1), FEL-NAP (1:2), FEL-NAP (2:1), and NTP-NAP (2:1) systems were compared with predicted *T*_g_ values calculated using the Fox Equation (1), which assumes ideal mixing [22,23]:1/*T_g_*^mix^ = *W*_1_/*T_g_*_1_ + *W*_2_/*T_g_*_2_
(1)
where *T*_g1_ and *T*_g2_ represent the glass transition temperatures of components 1 and 2, *W*_1_ and *W*_2_ are their respective weight fractions, and *T*_g_^mix^ is the predicted glass transition temperature of the coamorphous mixture. This equation assumes a linear relationship between the *T*_g_ values of the individual components (FEL, NTP, and NAP), with any deviations from the predicted values suggesting molecular interactions between the molecules.

The *T*_g_ values for pure amorphous forms of FEL, NTP, and NAP were 46.46 °C, 42.14 °C, and 6 °C, respectively (Table 5). Based on the Fox equation, the predicted *T*_g_ values for the coamorphous forms of FEL-NAP (1:1), FEL-NAP (1:2), FEL-NAP (2:1), and NTP-NAP (2:1) were 2.13 °C, 1.69 °C, 2.89 °C, and 2.80 °C, respectively. According to the Fox equation, if the experimental and predicted *T*_g_ values are similar, no molecular interaction occurs between the components. However, positive deviations from the predicted *T*_g_ values indicate molecular interactions between the components. The experimental *T*_g_ values for FEL-NAP (1:1), FEL-NAP (1:2), FEL-NAP (2:1), and NTP-NAP (2:1) of 18.62 °C, 23.25 °C, 13.28 °C, and 19.64 °C, respectively, were higher than the predicted *T*_g_ values, further confirming strong molecular interactions [24,25].

### 3.5. FT-IR Molecular Interaction Analysis

The FT-IR spectra are shown in Figure 6. In Figure 6A, the N-H stretching vibration (*V*_-NH_) of FEL was observed at 3460 cm^−1^, while the O-H stretching vibration (*V*_-OH_) of NAP appeared at 3188 cm^−1^. Additionally, the C=O stretching vibrations (*V*_C=O_) of the carboxylic acid ester in FEL and the carboxylic acid in NAP were both visible at 1699 cm^−1^. For FEL-NAP_PM_, peaks at 3371, 3199, and 1699 cm^−1^ indicated the presence of N-H, O-H, and C=O stretching vibrations with no noticeable shifts, suggesting no drug–drug interactions. However, in the coamorphous systems of FEL-NAP (1:1), FEL-NAP (1:2), and FEL-NAP (2:1), some peaks weakened or disappeared. Notably, the peak corresponding to the N-H stretching vibration (*V*_-NH_) at 3467 cm^−1^ shifted; the peak corresponding to the O-H stretching vibration (*V*_-OH_) disappeared; and the peak representing the stretching vibration of the carboxylic ester C=O bonds in FEL-NAP (1:1), FEL-NAP (1:2), and FEL-NAP (2:1) shifted from 1699 to 1689 cm^−1^, 1684 cm^−1^, and 1686 cm^−1^, respectively.

In Figure 6B, the N-H stretching vibration (*V*_-NH_) of NTP was observed at 3331 cm^−1^, while the O-H stretching vibration (*V*_-OH_) of NAP was again present at 3188 cm^−1^. The carboxylic acid ester C=O stretching vibrations (*V*_C=O_) for NTP and NAP appeared at 1689 cm^−1^ and 1684 cm^−1^, respectively. In NTP-NAP_PM_, peaks at 3315 cm^−1^ and 1649 cm^−1^ confirmed the presence of N-H and C=O stretching vibrations with no shifts, indicating no drug–drug interactions. However, in the coamorphous systems of NTP-NAP (1:1), NTP-NAP (1:2), and NTP-NAP (2:1), some peaks weakened or disappeared, with the N-H stretching vibration shifting from 3467 cm^−1^, and the O-H peak disappearing. Additionally, the C=O stretching vibrations shifted from 1689, 1684, and 1649 cm^−1^ to 1653 cm^−1^, suggesting molecular interactions between the components.

The quench-cooled systems showed the most significant peak shifts, particularly in the FEL-NAP and NTP-NAP systems (Figure 6C,D), strongly suggesting molecular interactions (specifically the potential location for hydrogen bonding) between the drugs. These hydrogen bonds can inhibit crystal nucleation and growth [26], lowering the Gibbs free energy of the coamorphous systems and increasing the nucleation energy barrier, thereby preventing crystallization of individual amorphous drugs. Furthermore, the formation of heterodimers between different drugs can delay recrystallization, enhancing the physical stability of the coamorphous systems. The strong intermolecular forces in these coamorphous forms also contribute to improved dissolution of poorly soluble drugs [27], while maintaining supersaturation [28].

Unlike systems that rely on proton transfer, the formation of coamorphous systems is driven by intermolecular forces, such as hydrogen bonding between the drugs, making these systems particularly advantageous. Overall, FT-IR analysis demonstrated clear molecular interactions between FEL and NAP, as well as between NTP and NAP, in the coamorphous systems. These interactions align with the observed deviations from the Fox equation, indicating stronger stabilization and improved dissolution in the test medium.

### 3.6. Dissolution Measurements and Calculation of Dissolution Rates

Figure 7A presents the dissolution profiles of crystalline FEL, FEL-NAP_PM_, amorphous FEL, and the FEL-NAP systems. As shown, only a small amount of crystalline FEL dissolved over 60 min, while FEL-NAP_PM_ demonstrated slightly improved dissolution. The FEL-NAP systems, particularly the 1:2 ratio, exhibited significantly higher dissolution compared to the other systems. Similarly, Figure 7B illustrates the dissolution profiles of crystalline NAP, FEL-NAP_PM_, and the FEL-NAP systems. Although crystalline NAP and FEL-NAP_PM_ displayed a noticeable amount of dissolution during the 60 min, the FEL-NAP quench-cooled systems, especially the NAP component, showed enhanced dissolution compared to both crystalline NAP and FEL-NAP_PM_. The dissolution of the FEL-NAP quench-cooled systems was markedly higher than that of crystalline FEL, crystalline NAP, and FEL-NAP_PM_, demonstrating the effectiveness of quench-cooling technology in enhancing the dissolution of BCS II drugs like FEL and NAP.

As shown in Appendix A, crystalline FEL had a relatively low dissolution rate (~0.062 μg/(mL·min)). However, the dissolution rates for FEL-NAP (1:1), FEL-NAP (1:2), and FEL-NAP (2:1) were higher than those for both crystalline FEL and FEL-NAP_PM_. Across various time points, the dissolution rates followed the order of FEL-NAP (1:2) > FEL-NAP (2:1) > FEL-NAP (1:1) > amorphous FEL > FEL-NAP_PM_ > crystalline FEL. Given that NAP is a representative RT-unstable amorphous drug (Appendix A), comparable dissolution rates were observed between crystalline NAP, FEL-NAP_PM_, and the FEL-NAP systems. In contrast, crystalline FEL exhibited a significantly lower dissolution rate. Overall, quench-cooling technology produced the greatest improvement in dissolution, as reflected in the dissolution curves and rates in different tested media.

Figure 7C,D display the dissolution profiles of crystalline NTP, crystalline NAP, NTP-NAP_PM_, amorphous NTP, and NTP-NAP quench-cooled systems in various media. Only small amounts of crystalline NTP and NAP dissolved over 60 min, with NTP-NAP_PM_ showing slight but insignificant improvements in dissolution. In contrast, the NTP-NAP quench-cooled systems exhibited significantly higher dissolution compared to crystalline FEL, crystalline NAP, and FEL-NAP_PM_, further demonstrating the potential of quench-cooling technology to significantly enhance the dissolution of BCS II drugs like NTP and NAP.

As shown in Appendix A, the dissolution rate of crystalline NTP was relatively low (~6.64 × 10^−3^ μg/(mL·min)). However, amorphous NTP and NTP-NAP (2:1) exhibited higher dissolution rates than crystalline NTP and NTP-NAP_PM_, which when ranked from greatest to least, followed the order of NTP-NAP (2:1) > amorphous NTP > NTP-NAP (1:2) > NTP-NAP (1:1) > NTP-NAP_PM_ > crystalline NTP. Additionally, the NTP-NAP systems exhibited higher dissolution rates for NAP compared to crystalline NAP and NTP-NAP_PM_, as shown in Appendix A. The significantly lower dissolution rates for crystalline NTP and NAP, compared to the quench-cooled systems, confirm that quench-cooling technology offers the most substantial improvement in dissolution across different media.

To better understand the dissolution behavior, the Noyes-Whitney equation (Equation (2)) was applied ([29,30,31]):dC/dT = KS(Cs − C) = Ds/vh(Cs − C)(2)
where dC/dT represents the dissolution rate (mg/(mL·min), *D* is the diffusion coefficient, *K* is the dissolution rate constant, *S* is the available surface area, *C_S_* is the saturated solubility of the drug in the medium, *v* is the selected volume of medium used for dissolution, *h* is the thickness of the diffusion layer, and *C* is the drug concentration in the bulk fluid at time *T*.

The Noyes–Whitney equation provides a theoretical framework for the dissolution of poorly soluble drugs. According to this equation, an increase in saturated solubility (*C_S_*) is the key factor driving the accelerated dissolution rate (dC/dT), thereby enhancing overall dissolution. The dissolution results confirm that the primary reason for the enhanced dissolution is the increase in *C_S_*. As demonstrated in the solubility assays, all amorphous systems exhibited significantly higher solubility compared to their crystalline counterparts and non-amorphous systems. Consistent with the findings in Section 3.7, the saturation solubility of the amorphous systems was markedly higher than that of non-amorphous formulations. Therefore, quench cooling technology effectively enhances dissolution by increasing the *C_S_* of poorly soluble drugs.

### 3.7. Saturation Solubility

As shown in Figure 8, the lowest solubility of FEL was 1.19 µg/mL across various media at 24 h. The FEL concentration in FEL-NAP_PM_ ranged from 1.74 to 2.33 µg/mL at 24 and 48 h, while the concentrations of amorphous FEL ranged from 4.43 to 4.66 µg/mL at the same time points, demonstrating a significant solubilization effect (*p* < 0.01). The solubility of FEL in the FEL-NAP (1:1), FEL-NAP (1:2), and FEL-NAP (2:1) systems increased significantly compared to crystalline FEL and FEL-NAP_PM_ (*p* < 0.001) at 24 and 48 h. Among these, the FEL-NAP (2:1) system exhibited the highest solubility (8.62–9.46 µg/mL) at both time points, showing a marked improvement over the other quench-cooled systems (*p* < 0.05).

Conversely, the lowest solubility of NAP was 23.68 µg/mL in various media at 48 h. In the FEL-NAP_PM_ system, NAP concentrations ranged from 30.74 to 43.76 µg/mL at 24 and 48 h, with a significant solubilization effect observed (*p* < 0.01). The solubility of NAP in FEL-NAP (1:1), FEL-NAP (1:2), and FEL-NAP (2:1) was substantially higher compared to crystalline NAP and FEL-NAP_PM_ (*p* < 0.001) at both time points. The FEL-NAP (2:1) system displayed the highest solubility for NAP (65.45–40.49 µg/mL), significantly outperforming the other FEL-NAP quench-cooled systems (*p* ˂ 0.05). Overall, the FEL-NAP systems, especially those produced using the 1:1 and 1:2 ratios, showed remarkable improvements in both dissolution and solubility compared to the FEL-NAP (2:1) system.

Figure 9 illustrates the solubility of NTP, with the lowest value of 0.12 µg/mL across various media at 24 h. In the NTP-NAP_PM_ system, NAP concentrations ranged from 0.65 to 0.96 µg/mL at 24 and 48 h, indicating a significant solubilization effect (*p* < 0.01). Compared to crystalline NTP and NTP-NAP_PM,_ the solubility of NTP in the NTP-NAP (1:1), NTP-NAP (1:2), and NTP-NAP (2:1) systems increased significantly (*p* < 0.001) at both time points. Among these, NTP-NAP (2:1) demonstrated the highest solubility (1.52–1.58 µg/mL), significantly higher than the other NTP-NAP quench-cooled systems (*p* ˂ 0.05).

For NAP, the lowest solubility in various media at 24 h was 59.01 µg/mL. The NAP concentration in NTP-NAP_PM_ ranged from 66.81 to 67.15 µg/mL at 24 and 48 h, showing a significant solubilization effect (*p* < 0.01). Compared to crystalline NTP and NTP-NAP_PM_, the solubility of NAP in the NTP-NAP (1:1), NTP-NAP (1:2), and NTP-NAP (2:1) systems increased significantly (*p* < 0.001) at both time points. The NTP-NAP (1:2) system exhibited the highest NAP solubility (104.14–102.29 µg/mL), significantly outperforming the other NTP-NAP quench-cooled systems (*p* ˂ 0.05).

The incorporation of NAP in the quench-cooled systems was aimed at promoting NTP dissolution. The quench-cooled NTP-NAP systems demonstrated superior dissolution and solubility, underscoring the benefits of this approach. Quench-cooling technology enables FEL, NTP, and NAP to fully mix in the molten state, ensuring that the majority of the drug exists in an amorphous state. During amorphization, the drugs transition from an ordered crystalline state to a disordered glassy state within the paired drug matrix, resulting in significant solubility improvements.

In the quench-cooled systems, the transition of drugs from an ordered crystalline state to a disordered glassy state (Δ*G*_lattice_ = 0) enhances solubility due to the higher energy state of the amorphous form. The dissolution free energy involves three components: Δ*G*_solution_, Δ*G*_lattice_, and Δ*G*_solvation_, defined by the following relationship:Δ*G*_solution_ = Δ*G*_lattice_ + Δ*G*_solvation_(3)
where Δ*G*_solution_ represents the Gibbs free energy of the solubilization process, Δ*G*_lattice_ represents the lattice energy, and Δ*G*_solvation_ represents the solvation energy. The dissolution process involves three steps: disruption of intermolecular connections within the solid drug (solute–solute), disruption of solvent intermolecular connections (solvent–solvent), and the formation of new molecular interactions between solute and solvent molecules (solute–solvent). The smaller the Δ*G*_solution_, the lower the energy barrier for dissolution and the greater the solubility.

In ideal solutions, Δ*G*_solution_ is primarily determined by Δ*G*_lattice_ [32], meaning that the higher the lattice energy, the lower the solubility. However, in non-ideal solutions, when Δ*G*_solution_ ≥ Δ*G*_lattice_ (Δ*G*_solution_ ≤ 0), the dissolution process proceeds spontaneously. To improve drug solubility, both Δ*G*_lattice_ and Δ*G*_solvation_ must be minimized.

In this study, we enhanced solubility by altering Δ*G*_lattice_ using coamphorous systems that alter the molecular arrangement of the drugs. In such systems, atoms are arranged in a disordered manner over long distances but maintain some localized order at short distances, resembling the structure of a supercooled liquid or glass, with no internal crystalline lattice (Δ*G*_lattice_ = 0) and a thermodynamically high-energy state [33,34]. Consequently, the energy required to disrupt the intermolecular forces of coamorphous drugs is greatly reduced, significantly improving dissolution and solubility of a drug. Given that coamorphous systems have higher free energy than their crystalline counterparts, they dissolve more easily in water and reach equilibrium solubility more rapidly.

### 3.8. PXRD Assessment of the Long-Term Stability of the Amorphous State

The long-term stability of the amorphous state was evaluated using PXRD (Figure 10). As shown in Figure 10A–F, no distinct crystalline diffraction peaks were observed in the samples shortly after preparation, confirming that all samples were initially in an amorphous state. However, by comparing the data at 0 months with later time points, it became clear that after 1 month (Figure 10A1), crystalline diffraction peaks appeared at 10.6°, 16.68°, 20.88°, 23.64°, 26.8°, and 29.84°, indicating that pure amorphous FEL without crystallization inhibitors was unstable and had begun to crystallize during the first month of storage.

While FEL-NAP (1:1) exhibited prominent crystal diffraction peaks after 1 month (Figure 10B1), only a few such peaks were observed in the FEL-NAP (1:2) (Figure 10C1) and FEL-NAP (2:1) (Figure 10D1) systems, suggesting that crystallization took place only in the FEL-NAP (1:1) formulation. For amorphous NTP, the diffraction pattern after 1 month (Figure 10E1) remained similar to that at 0 months, except for the NTP-NAP (2:1) system (Figure 10F1), which displayed clear diffraction peaks, indicating crystallization had occurred. Both the FEL-NAP and NTP-NAP formulations showed comparable ability to maintain their amorphous state during storage, in contrast to findings from previous studies [35,36,37].

Overall, the ddCAMs FEL-NAP (1:2) and FEL-NAP (2:1) prepared by quench cooling demonstrated greater physical stability compared to traditional formulations. These results further illustrate that the physical stability of ddCAMs is influenced by the drug ratios, which play a critical role in maintaining the stability of amorphous NAP at RT. These findings offer valuable insights for the design and optimization of ddCAMs, particularly when incorporating components prone to RT instability.

### 3.9. “Spring” and “Parachute” Parameters

Amorphous solids often induce a “spring–parachute” process in solution, consisting of the “spring” phase, which is driven by amorphization, and the “parachute” phase, where crystallization inhibitors (such as polymers or drugs) prevent recrystallization. The height of the “spring” phase is crucial for enhancing dissolution and potentially improving absorption in vivo [38]. In contrast, the “parachute” phase helps maintain the stability of the amorphous state for an extended period [39]. The initial dissolution level is closely associated with the “spring” effect, while the final solubility reflects the “parachute” effect [40,41]. To assess these parameters in quench-cooled systems compared to amorphous and crystalline systems, we defined the dissolution ratio at 60 min as the “spring” parameter (based on ”spring” height) and the solubility ratio at 24 and 48 h as the “parachute” parameter.

Table 6 and Table 7 present the “spring” and “parachute” parameters for the FEL-NAP and NTP-NAP systems at 1, 24, and 48 h. The dissolution ratios of the FEL-NAP/amorphous FEL and NTP-NAP/amorphous NTP systems showed significant increases (*p* < 0.001 or *p* < 0.01) at 60 min. As indicated in Table 6, ^sp^Ratio_F-N(1:1)/aFEL_, ^sp^Ratio_F-N(1:2)/aFEL_, and ^sp^Ratio_F-N(2:1)/aFEL_ were 1.27 ± 0.64, 1.72 ± 0.85, and 1.53 ± 0.46, respectively, indicating similar results across the FEL-NAP systems. However, the ^sp^Ratio_N-N(2:1)/aNTP_ was higher (1.61 ± 0.63) compared to ^sp^Ratio_N-N(1:1)/aNTP_ (0.69 ± 0.30) and ^sp^Ratio_N-N(1:2)/aNTP_ (0.84 ± 0.14). These findings demonstrate higher “spring” parameters for systems such as ^sp^Ratio_F-N(1:1)/aFEL_, ^sp^Ratio_F-N(1:2)/aFEL_, ^sp^Ratio_F-N(2:1)/aFEL_, and ^sp^Ratio_N-N(2:1)/aNTP_, reflecting enhanced solubility provided by amorphization.

For the “parachute” parameters (Table 6), ^pa^Ratio_F-N(1:1)/aFEL_, ^pa^Ratio_F-N(1:2)/aFEL_, and ^pa^Ratio_F-N(2:1)/aFEL_ remained comparable at 24 h (1.49 ± 0.42, 1.72 ± 0.44, and 1.95 ± 0.79, respectively) and 48 h (1.61 ± 0.29, 1.75 ± 0.48, and 2.03 ± 0.75, respectively). In contrast, values for ^pa^Ratio_N-N(2:1)/aNTP_ (1.25 ± 0.081 at 24 h and 1.66 ± 0.12 at 48 h) exceeded those of ^pa^Ratio_N-N(1:1)/aNTP_ (0.89 ± 0.088 at 24 h and 1.33 ± 0.070 at 48 h) and ^pa^Ratio_N-N(1:2)/aNTP_ (0.94 ± 0.077 at 24 h and 1.46 ± 0.080 at 48 h). The higher “parachute” parameters in the coamorphous systems, such as ^pa^Ratio_F-N(1:1)/aFEL_, ^pa^Ratio_F-N(1:2)/aFEL_, ^pa^Ratio_F-N(2:1)/aFEL_, and ^pa^Ratio_N-N(2:1)/aNTP_, demonstrate that quench-cooling significantly enhanced the stability of the pure amorphous forms of insoluble drugs in ddCAM systems, promoting longer supersaturation.

The supersaturation state was characterized by two factors: the supersaturation ratio *S* (Equation (4)) and the relative supersaturation index σ (Equation (5)):S = C_h_/C(4)
σ = S − 1 = (C_h_ − C)/C(5)
where *C_h_* represents the concentration at time point *T*, and *C* is the amorphous or crystalline solubility. When *σ* < 0 (*S* < 1), the solution is unsaturated; when *σ* < 0 (*S* < 1), the solution is saturated; when *σ* > 0 (*S* > 1), the solution is supersaturated [42]. Kinetic solubility or supersaturation profiles were used to assess solution stability and the duration of drug supersaturation before recrystallization. “Spring” parameters such as ^sp^Ratio_F-N(1:1)/aFEL_, ^sp^Ratio_F-N(1:2)/aFEL_, ^sp^Ratio_F-N(2:1)/aFEL_, and ^sp^Ratio_N-N(2:1)/aNTP_ exceeded 1 (C_h_/C > 1), confirming that the coamorphous forms generated supersaturated solutions. Meanwhile, the parachute parameters for these systems remained stable after 24 and 48 h, with the maximum supersaturation level for FEL-NAP (2:1) more than twice that of the amorphous FEL system (based on the value of ^pa^Ratio_F-N(2:1)/aFEL_), indicating that coamorphous systems maintained a higher level of supersaturation.

Analysis of both the solubility and “parachute” parameters demonstrated that coamorphous systems such as FEL-NAP (1:1), FEL-NAP (1:2), FEL-NAP (2:1), and NTP-NAP (2:1) enhanced solubility and maintained solution stability of both drugs for extended periods, achieving higher drug concentrations in line with supersaturation theory. No significant solubility differences were found between crystalline and amorphous NAP at 24 and 48 h, likely due to the absence of crystal inhibitors (paired drugs).

These findings collectively demonstrate that coamorphous systems provided superior stability, dissolution enhancement, and crystallization inhibition compared to corresponding amorphous forms. Quench-cooled coamorphous systems exhibited a pronounced “spring” effect, significantly improving initial dissolution, while the maintenance of supersaturation (“parachute” effect) was strongly dependent on the drug ratios and pairings in the ddCAM formulations, particularly when they contained an RT-unstable component. Quench-cooling technology enhanced both the “spring” and “parachute” parameters, with coamorphous systems being more effective than non-coamorphous systems in improving the dissolution of FEL and NTP. Coamorphous systems also exhibited superior stability by delaying crystallization and enhancing the maintenance of the supersaturated state.

Following classical nucleation theory [43,44], the crystallization profile can be described by Equation (6):(6)$$J=A\;\exp\frac{16\pi \gamma^{3}v^{2}}{3k^{3}T^{3}(\ln S)}$$
where *T* is the experimental temperature, *v* is the molecular volume, *k* is the Boltzmann constant, *γ* is the interfacial tension, and *S* is the degree of supersaturation (i.e., the ratio of the solution concentration to the solubility of the amorphous or crystalline form). According to Equation (6), crystallization is influenced by thermodynamic driving forces, as well as various factors affecting the nucleation rate (*J*), with *J* depending on the degree of supersaturation (*S*). Thus, amorphous–crystalline systems with higher solubility ratios are expected to nucleate and precipitate more readily. In this study, amorphous–crystalline systems demonstrated a tendency to nucleate and precipitate from solution (Table 6 and Table 7), with amorphous FEL and NTP reaching higher concentrations in solution than their crystalline counterparts. Crystallization inhibition was evident, as indicated by the lower recrystallization rate. Issues related to physical stability (such as amorphous–amorphous phase separation, nucleation, and crystallization of the API) during storage are recognized as key challenges [45,46].

Effective maintenance of supersaturation through crystallization inhibition is a crucial feature of well-designed coamorphous systems. To select optimal drug ratios for these systems, various ratios of FEL-NAP and NTP-NAP in their supersaturated states were evaluated. Enhanced intermolecular interactions induced by coamorphization were confirmed via FT-IR analysis and supported by *T_g_* analysis (DSC), PXRD stability assessments, and the demonstrated ability of drugs in these ratios to form stable coamorphous states. The stronger intermolecular interaction within coamorphous systems, compared to their crystalline and amorphous counterparts, played a pivotal role in inhibiting crystallization. Hydrogen bonding between FEL-NAP and NTP-NAP likely contributed to maintaining the supersaturated state in solution. Thus, the strategic design of coamorphous systems not only enhances dissolution but also ensures prolonged supersaturation, offering a promising approach for improving the bioavailability of poorly soluble drugs.

## 4. Conclusions

The study focused on model BCS II drug pairs “FEL-NAP” and “NTP-NAP” ddCAMs. The physicochemical properties of various quench-cooled systems were characterized using SEM, PXRD, and DSC. The results confirmed successful fabrication of ddCAMs for most ratios, which exhibited significantly enhanced dissolution and solubility compared to their non-coamorphous counterparts. Enhanced intermolecular interactions were observed through FT-IR analysis, suggesting reduced molecular mobility in the amorphous forms of the ddCAMs. This reduction in mobility likely impedes intermolecular binding, thereby inhibiting transcrystallization. Experimentally, the “spring” parameters for FEL-NAP (1:1), FEL-NAP (1:2), FEL-NAP (2:1), and NTP-NAP (2:1) were found to be relatively high during the initial stage, suggesting their potential to enhance drug absorption in vivo. In the later stages, the “parachute” parameters of these amorphous ddCAMs were also elevated, playing a crucial role in maintaining the amorphous state over an extended period. All coamorphous systems demonstrated improved long-term stability of the amorphous state at RT compared with pure amorphous NAP. Coamorphous or amorphous drugs generally exhibited much higher solubility than their crystalline forms. Our findings provide deeper insight into the co-amorphization, dissolution, and stability characteristics of specific drug–drug coamorphous systems, offering valuable guidance for developing new coamorphous formulations incorporating FEL-NAP and NTP-NAP and other RT-unstable drugs. On the other hand, by preparing different representative ratios between NAP and (FEL and NTP) for various studies, theoretical guidance and data support have been provided for further clinical combination therapy based on these drug–drug pairs. The actual dosage of drug–drug in clinical practice still needs to be determined and regulated according to patients’ condition, and this will be another focus in our future work.

## Figures and Tables

**Figure 1 pharmaceutics-16-01488-f001:**
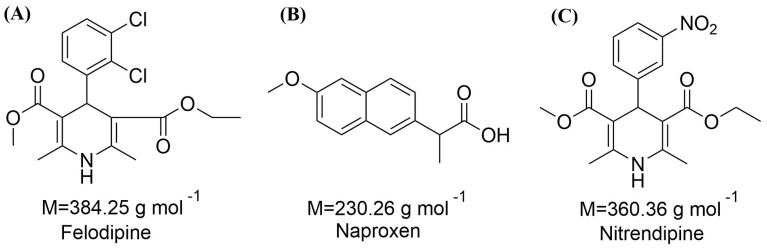
Chemical structures of (**A**) felodipine (FEL); (**B**) naproxen (NAP); (**C**) nitrendipine (NTP).

**Figure 2 pharmaceutics-16-01488-f002:**
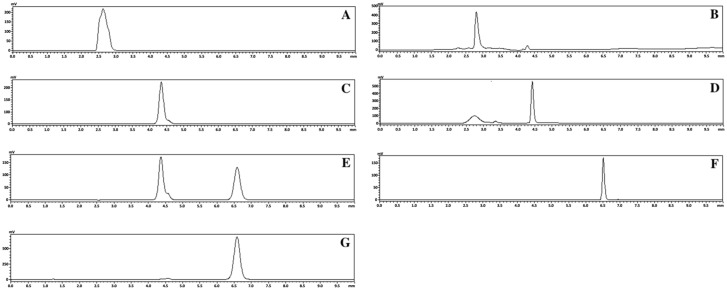
HPLC chromatogram results for crystalline FEL (**A**), FEL-related samples of FEL-NAP (1:1) (**B**), crystalline NAP (**C**), NAP-related samples of FEL-NAP (1:1) (**D**), NAP-related samples of NTP-NAP (1:1) (**E**), crystalline NTP (**F**), and NTP-related samples of NTP-NAP (1:1) (**G**).

**Figure 3 pharmaceutics-16-01488-f003:**
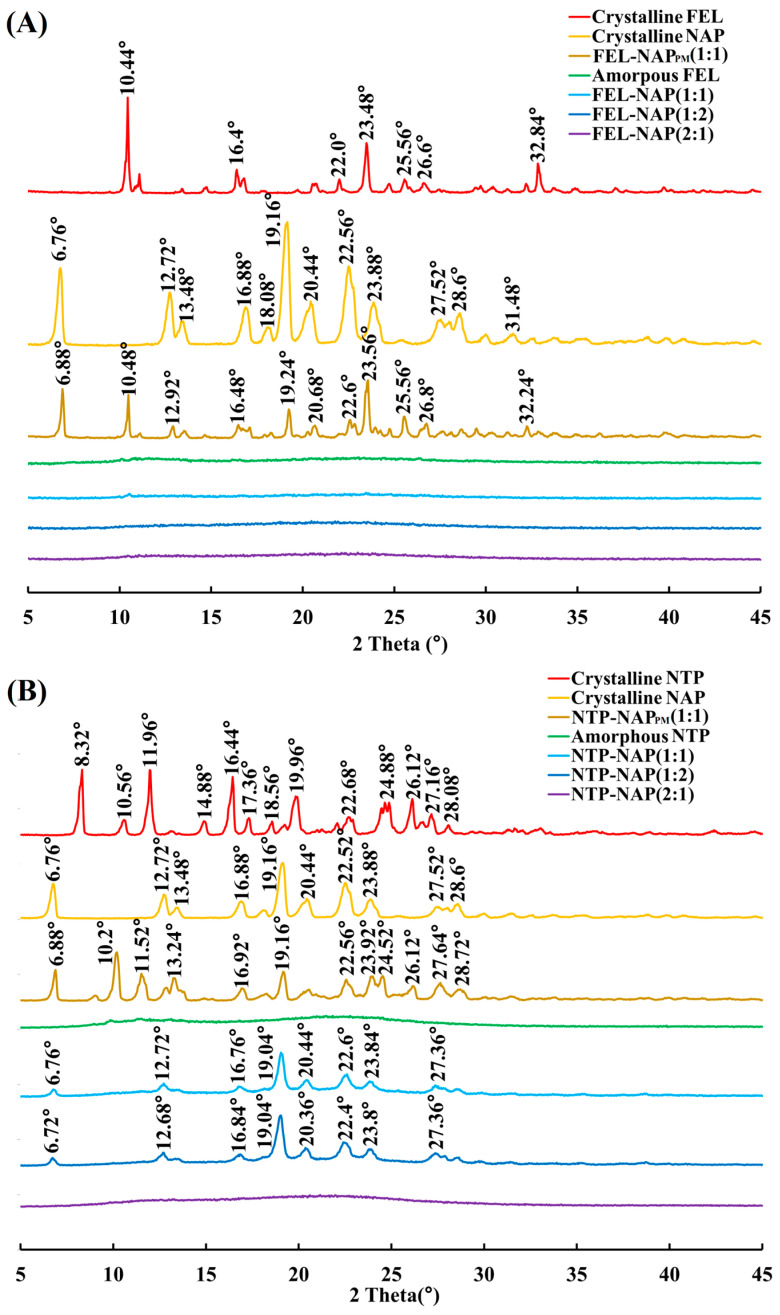
(**A**) PXRD results for crystalline FEL, crystalline NAP, FEL-NAP_PM_ (1:1), amorphous FEL, FEL-NAP (1:1), FEL-NAP (1:2), and FEL-NAP (2:1); (**B**) PXRD results for crystalline NTP, crystalline NAP, NTP-NAP_PM_ (1:1), amorphous NTP, NTP-NAP (1:1), NTP-NAP (1:2), and NTP-NAP (2:1).

**Figure 4 pharmaceutics-16-01488-f004:**
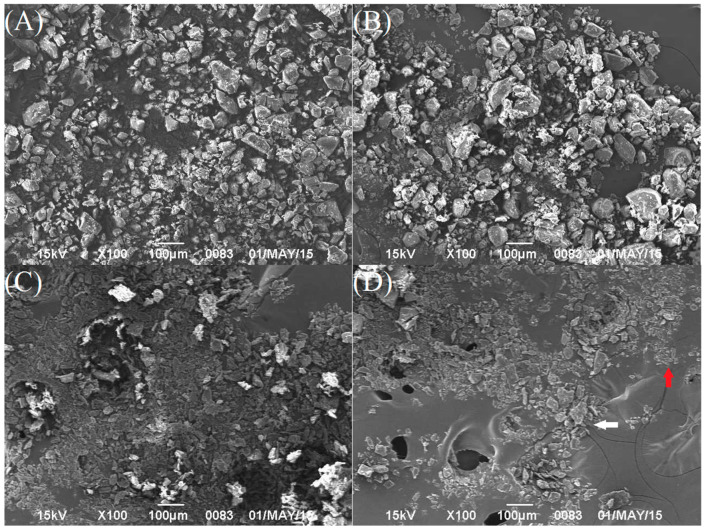
SEM images of (**A**) crystalline FEL; (**B**) crystalline NTP; (**C**) crystalline NAP; (**D**) FEL-NAP_PM_ (1:1) (FEL: white arrow and NAP: red arrow); (**E**) FEL-NAP (1:1); (**F**) the physical mixture NTP-NAP_PM_ (1:1) (NTP: green arrow and NAP: red arrow); (**G**) NTP-NAP (2:1).

**Figure 5 pharmaceutics-16-01488-f005:**
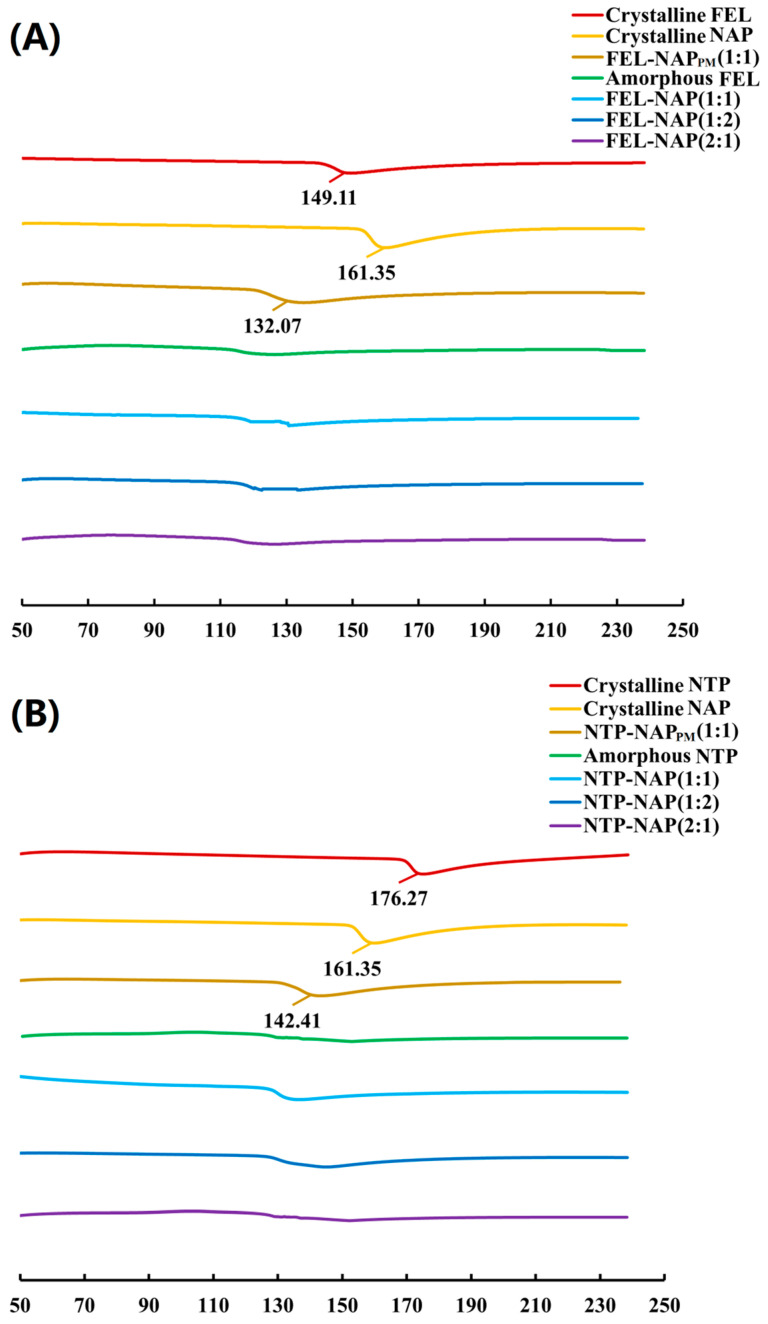
(**A**) DSC thermograms of all samples, including crystalline FEL, crystalline NAP, FEL-NAP_PM_ (1:1), amorphous FEL, FEL-NAP (1:1), FEL-NAP (1:2), and FEL-NAP (2:1); (**B**) DSC thermograms of all samples including crystalline NTP, crystalline NAP, NTP-NAP_PM_ (1:1), amorphous NTP, NTP-NAP (1:1), NTP-NAP (1:2), and NTP-NAP (2:1).

**Figure 6 pharmaceutics-16-01488-f006:**
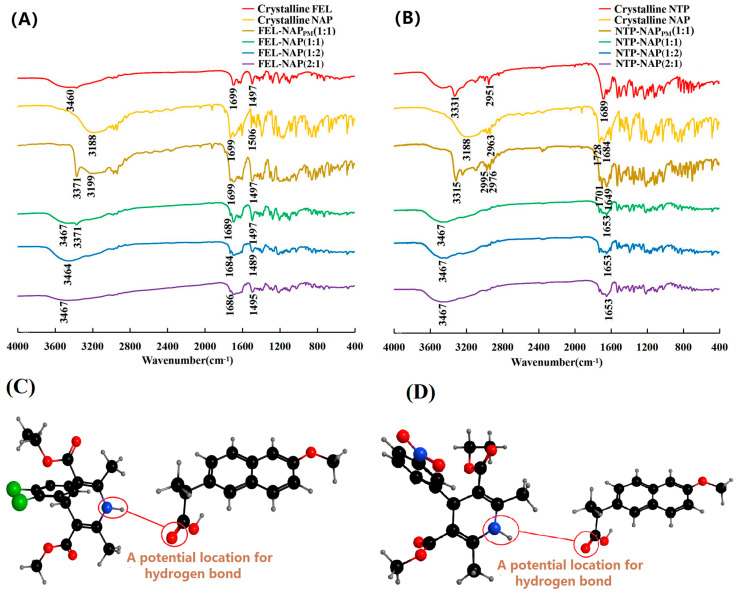
(**A**) FT-IR spectra of crystalline FEL, crystalline NAP, FEL-NAP_PM_ (1:1), FEL-NAP (1:1), FEL-NAP (1:2), and FEL-NAP (2:1); (**B**) FT-IR spectra of crystalline NTP, crystalline NAP, NTP-NAP_PM_ (1:1), NTP-NAP (1:1), NTP-NAP (1:2), and NTP-NAP (2:1); (**C**,**D**) Spatial structures (black balls represent carbon atoms, grey balls represent hydrogen atoms, red balls represent oxygen atoms, blue balls represent nitrogen atoms, and green balls represent chlorine atoms) of FEL-NAP (**C**) and NTP-NAP (**D**).

**Figure 7 pharmaceutics-16-01488-f007:**
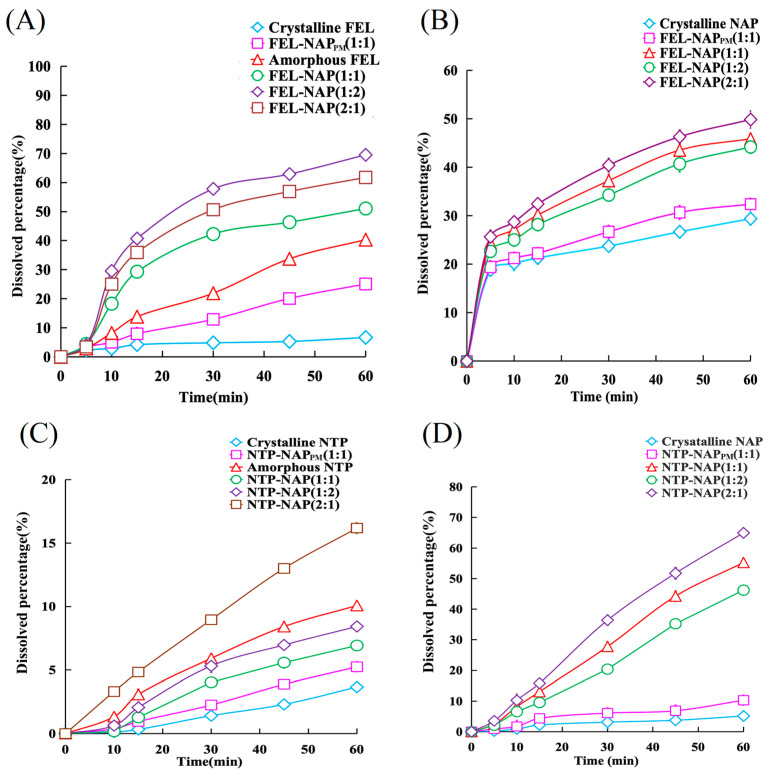
(**A**) Dissolution profiles of FEL-related samples containing equivalent amounts of FEL (50 mg), including FEL-NAP (1:1, 1:2, and 2:1), amorphous FEL, FEL-NAP_PM_ (1:1), and crystalline FEL, in 900 mL of 0.1 M hydrochloric acid (37 °C). (**B**) Dissolution profiles of NAP-related samples containing equivalent amounts of NAP (50 mg), including FEL-NAP (1:1, 1:2, and 2:1), FEL-NAP_PM_ (1:1), and crystalline NAP, in 900 mL of 0.25% SDS solution (pH = 7.2, 37 °C). (**C**) Dissolution profiles of NTP-related samples containing equivalent amounts of NTP (50 mg), including NTP-NAP (1:1, 1:2, and 2:1), amorphous NTP, NTP-NAP_PM_ (1:1), and crystalline NTP, in 900 mL of 0.1 M hydrochloric acid solution (37 °C). (**D**) Dissolution profiles of NAP-related samples containing equivalent amounts of NAP (50 mg), including NTP-NAP (1:1, 1:2, and 2:1), NTP-NAP_PM_ (1:1), and crystalline NAP, in 900 mL of 0.25% SDS solution (pH = 7.2, 37 °C). Error bars represent the standard deviation, n = 3.

**Figure 8 pharmaceutics-16-01488-f008:**
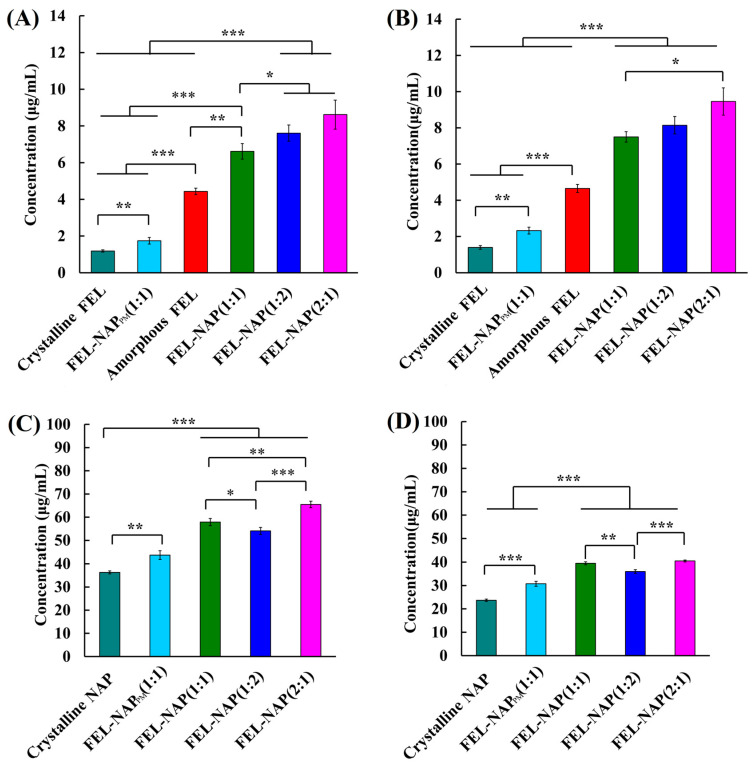
(**A**,**B**) Solubility of FEL-related samples including FEL-NAP (1:1, 1:2, and 2:1), amorphous FEL, FEL-NAP_PM_ (1:1), and crystalline FEL in 900 mL H_2_O at 37 °C for 24 h (**A**) and 48 h (**B**). (**C**,**D**) Solubility of NAP-related samples including FEL-NAP (1:1, 1:2, and 2:1), FEL-NAP_PM_ (1:1), and crystalline NAP in 900 mL H_2_O at 37 °C for 24 h (**C**) and 48 h (**D**). Error bars represent the standard deviation, n = 3; * *p* < 0.05, ** *p* < 0.01, *** *p* < 0.001.

**Figure 9 pharmaceutics-16-01488-f009:**
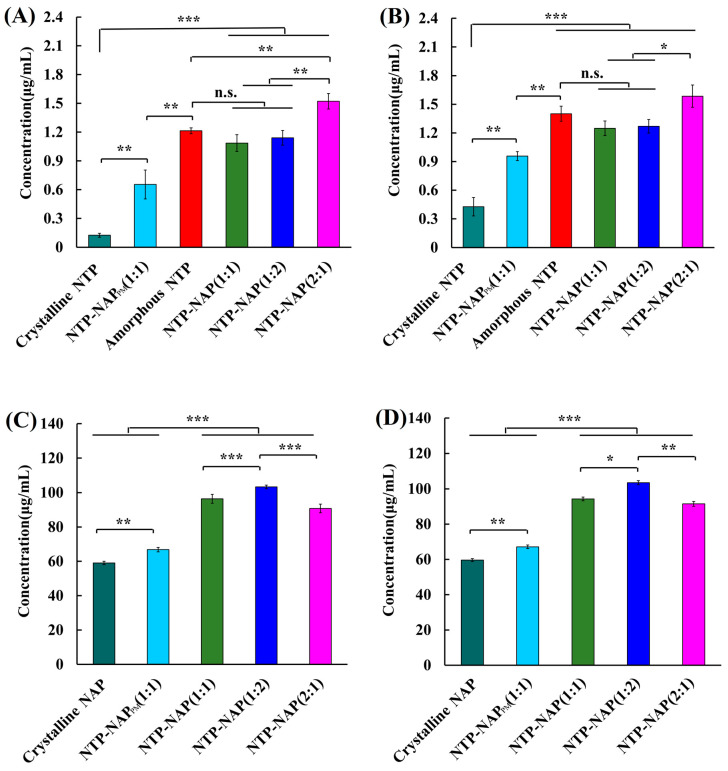
(**A**,**B**): Solubility of NTP-related samples including NTP-NAP (1:1, 1:2, and 2:1), amorphous NTP, NTP-NAP_PM_ (1:1), and crystalline NTP in 900 mL H_2_O at 37 °C for 24 h (**A**) and 48 h (**B**). (**C**,**D**) Solubility of NAP-related samples including NTP-NAP (1:1, 1:2, and 2:1), NTP-NAP_PM_ (1:1), and crystalline NAP in 900 mL H_2_O at 37 °C for 24 h (**C**) and 48 h (**D**). Error bars represent the standard deviation, n = 3; n.s. (No significance), * *p* < 0.05, ** *p* < 0.01, *** *p* < 0.001.

**Figure 10 pharmaceutics-16-01488-f010:**
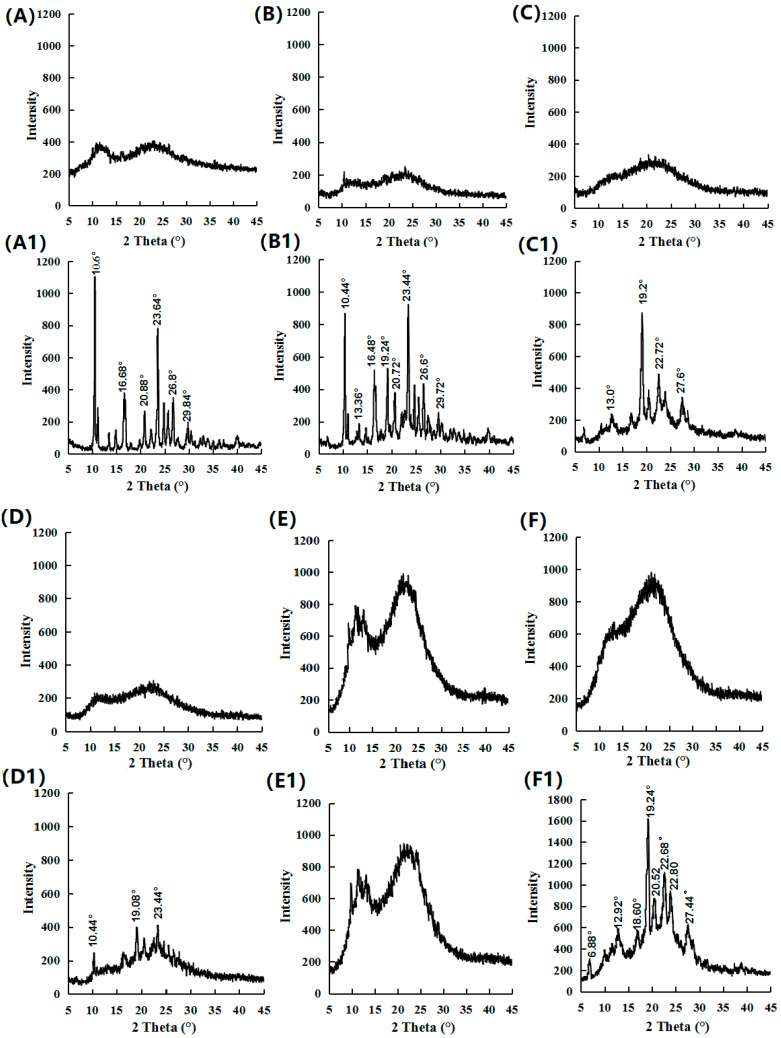
PXRD results for (**A**) amorphous FEL (0 month), (**A1**) amorphous FEL (1 month), (**B**) FEL-NAP (1:1) (0 month), (**B1**) FEL-NAP (1:1) (1 month), (**C**) FEL-NAP (1:2) (0 month), (**C1**) FEL-NAP (1:2) (1 month), (**D**) FEL-NAP (2:1) (0 month), (**D1**) FEL-NAP (2:1) (1 month), (**E**) amorphous NTP (0 month), (**E1**) amorphous NTP (1 month), (**F**) NTP-NAP (2:1) (0 month), and (**F1**) NTP-NAP (2:1) (1 month).

**Table 1 pharmaceutics-16-01488-t001:** Overview of samples analyzed in this work (including molecular weights and abbreviations used).

Samples	Abbreviation	Molecular Weight (g/mol)
Felodipine	FEL	384.25
Nitrendipine	NTP	360.36
Naproxen	NAP	230.26
FEL-NAP physical mixtures	FEL-NAP_PM_	N/A (Not Available)
NTP-NAP physical mixtures	NTP-NAP_PM_	N/A
FEL-NAP quench cooling system (1:1)	FEL-NAP (1:1)	N/A
FEL-NAP quench cooling system (1:2)	FEL-NAP (1:2)	N/A
FEL-NAP quench cooling system (2:1)	FEL-NAP (2:1)	N/A
NTP-NAP quench cooling system (1:1)	NTP-NAP (1:1)	N/A
NTP-NAP quench cooling system (1:2)	NTP-NAP (1:2)	N/A
NTP-NAP quench cooling system (2:1)	NTP-NAP (2:1)	N/A

**Table 2 pharmaceutics-16-01488-t002:** Analysis of FEL, NTP, and NAP.

Samples	The Mobile Phase			The Detection Wavelength (nm)	The Calibration Range
	A	B	A:B (*v*/*v*)		
FEL	Methanol	0.05 mM acetic acid	90:10	361	1~60 (μg/mL)
NAP	Methanol	0.05 mM acetic acid	80:20	317	0.01~0.30 (mg/mL)
NTP	Methanol	0.05 mM acetic acid	70:30	353	1~50 (μg/mL)

**Table 3 pharmaceutics-16-01488-t003:** FEL and NAP content determination results of different quench-cooling system samples by HPLC assay (mean ± SD, n = 3).

Samples	Content of FEL (%)	Content of NAP (%)
FEL-NAP (1:1)	98.29 ± 2.32	98.89 ± 0.53
FEL-NAP (1:2)	98.61 ± 1.16	99.41 ± 1.25
FEL-NAP (2:1)	99.25 ± 0.66	99.71 ± 1.01

**Table 4 pharmaceutics-16-01488-t004:** NTP and NAP content determination results of different quench-cooling system samples by HPLC assay (mean ± SD, n = 3).

Samples	Content of NTP (%)	Content of NAP (%)
NTP-NAP (1:1)	98.90 ± 1.17	96.10 ± 1.22
NTP-NAP (1:2)	97.40 ± 1.13	96.01 ± 1.24
NTP-NAP (2:1)	99.20 ± 0.92	96.27 ± 0.28

**Table 5 pharmaceutics-16-01488-t005:** Experimental *T*_g_ and predicted *T*_g_ of amorphous FEL; amorphous NTP; amorphous NAP; and coamorphous systems including FEL-NAP (1:1), FEL-NAP (1:2), FEL-NAP (2:1), and NTP-NAP (2:1) determined by DSC.

Samples	Experimental *T_g_* (°C)	Predicted *T_g_* (°C)
Amorphous FEL	46.46	--
Amorphous NTP	42.14	--
Amorphous NAP	6	--
FEL-NAP (1:1)	18.62	2.13
FEL-NAP (1:2)	23.25	1.69
FEL-NAP (2:1)	13.28	2.89
NTP-NAP (2:1)	19.64	2.80

**Table 6 pharmaceutics-16-01488-t006:** The “spring” parameter is defined by the dissolution ratio of the quench-cooled systems to the corresponding amorphous or crystalline systems at 1 h. Statistical analysis was carried out between the quench-cooled systems and corresponding amorphous (or crystalline) drugs through unpaired Student’s *t*-test (** *p* < 0.01, *** *p* < 0.001 for the mean ± SD, n = 3).

Dissolution Ratios	Abbreviations	“Spring” Parameters
FEL-NAP(1:1)/amorphous FEL	^sp^Ratio_F-N(1:1)/aFEL_	1.27 ± 0.64
FEL-NAP(1:2)/amorphous FEL	^sp^Ratio_F-N(1:2)/aFEL_	1.72 ± 0.85
FEL-NAP(2:1)/amorphous FEL	^sp^Ratio_F-N(2:1)/aFEL_	1.53 ± 0.46
NTP-NAP(1:1)/amorphous NTP	^sp^Ratio_N-N(1:1)/aNTP_	0.69 ± 0.30
NTP-NAP(1:2)/amorphous NTP	^sp^Ratio_N-N(1:2)/aNTP_	0.84 ± 0.14
NTP-NAP(2:1)/amorphous NTP	^sp^Ratio_N-N(2:1)/aNTP_	1.61 ± 0.63
FEL-NAP(1:1)/crystalline FEL	^sp^Ratio_F-N(1:1)/cFEL_	7.64 ± 1.55 ***
FEL-NAP(1:2)/crystalline FEL	^sp^Ratio_F-N(1:2)/cFEL_	10.41 ± 1.37 ***
FEL-NAP(2:1)/crystalline FEL	^sp^Ratio_F-N(2:1)/cFEL_	9.24 ± 0.84 ***
NTP-NAP(1:1)/crystalline NTP	^sp^Ratio_N-N(1:1)/cNTP_	1.89 ± 1.00 **
NTP-NAP(1:2)/crystalline NTP	^sp^Ratio_N-N(1:2)/cNTP_	2.30 ± 0.46 **
NTP-NAP(2:1)/crystalline NTP	^sp^Ratio_N-N(2:1)/cNTP_	4.42 ± 0.85 **
FEL-NAP(1:1)/crystalline NAP	^sp^Ratio_F-N(1:1)/cNAP_	1.56 ± 0.12
FEL-NAP(1:2)/crystalline NAP	^sp^Ratio_F-N(1:2)/cNAP_	1.70 ± 0.79
FEL-NAP(2:1)/crystalline NAP	^sp^Ratio_F-N(2:1)/cNAP_	1.50 ± 0.35
NTP-NAP(1:1)/crystalline NAP	^sp^Ratio_N-N(1:1)/cNAP_	10.73 ± 0.60 ***
NTP-NAP(1:2)/crystalline NAP	^sp^Ratio_N-N(1:2)/cNAP_	12.62 ± 0.41 ***
NTP-NAP(2:1)/crystalline NAP	^sp^Ratio_N-N(2:1)/cNAP_	8.98 ± 0.72 ***

**Table 7 pharmaceutics-16-01488-t007:** The “parachute” parameter, defined as the solubility ratio of the quench-cooled systems to the corresponding amorphous or crystalline systems at 24 and 48 h. Statistical analysis was carried out between the quench-cooled systems and corresponding amorphous (or crystalline) drugs through unpaired Student’s *t*-test (** *p* < 0.01, *** *p* < 0.001 for the mean ± SD, n = 3).

Solubility Ratios	Abbreviations	“Parachute” Parameters	
		24 h	48 h
FEL-NAP(1:1)/amorphous FEL	^pa^Ratio_F-N(1:1)/aFEL_	1.49 ± 0.42	1.61 ± 0.29
FEL-NAP(1:2)/amorphous FEL	^pa^Ratio_F-N(1:2)/aFEL_	1.72 ± 0.44	1.75 ± 0.48
FEL-NAP(2:1)/amorphous FEL	^pa^Ratio_F-N(2:1)/aFEL_	1.95 ± 0.79	2.03 ± 0.75
NTP-NAP(1:1)/amorphous NTP	^pa^Ratio_N-N(1:1)/aNTP_	0.89 ± 0.088	1.33 ± 0.070
NTP-NAP(1:2)/amorphous NTP	^pa^Ratio_N-N(1:2)/aNTP_	0.94 ± 0.077	1.46 ± 0.080
NTP-NAP(2:1)/amorphous NTP	^pa^Ratio_N-N(2:1)/aNTP_	1.25 ± 0.081	1.66 ± 0.12
FEL-NAP(1:1)/crystalline FEL	^pa^Ratio_F-N(1:1)/cFEL_	5.56 ± 0.66 ***	5.38 ± 0.85 ***
FEL-NAP(1:2)/crystalline FEL	^pa^Ratio_F-N(1:2)/cFEL_	6.40 ± 0.74 ***	5.84 ± 0.81 ***
FEL-NAP(2:1)/crystalline FEL	^pa^Ratio_F-N(2:1)/cFEL_	7.24 ± 1.77 ***	6.78 ± 0.48 ***
NTP-NAP(1:1)/crystalline NTP	^pa^Ratio_N-N(1:1)/cNTP_	8.70 ± 0.37 ***	2.97 ± 0.73 **
NTP-NAP(1:2)/crystalline NTP	^pa^Ratio_N-N(1:2)/cNTP_	9.14 ± 0.84 ***	3.28 ± 0.82 **
NTP-NAP(2:1)/crystalline NTP	^pa^Ratio_N-N(2:1)/cNTP_	12.18 ± 1.03 ***	3.71 ± 1.00 **
FEL-NAP(1:1)/crystalline NAP	^pa^Ratio_F-N(1:1)/cNAP_	1.60 ± 0.15	1.67 ± 0.065
FEL-NAP(1:2)/crystalline NAP	^pa^Ratio_F-N(1:2)/cNAP_	1.49 ± 0.15	1.52 ± 0.077
FEL-NAP(2:1)/crystalline NAP	^pa^Ratio_F-N(2:1)/cNAP_	1.80 ± 0.14	1.71 ± 0.040
NTP-NAP(1:1)/crystalline NAP	^pa^Ratio_N-N(1:1)/cNAP_	1.63 ± 0.26	1.58 ± 0.11
NTP-NAP(1:2)/crystalline NAP	^pa^Ratio_N-N(1:2)/cNAP_	1.75 ± 0.10	1.74 ± 0.11
NTP-NAP(2:1)/crystalline NAP	^pa^Ratio_N-N(2:1)/cNAP_	1.54 ± 0.25	1.53 ± 0.13

## Data Availability

Data supporting reported results can be found.

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
