# Peer review of "Co-Amorphization, Dissolution, and Stability of Quench-Cooled Drug–Drug Coamorphous Supersaturating Delivery Systems with RT-Unstable Amorphous Components"

_pharmaceutics, 2024, doi:10.3390/pharmaceutics16121488_

Round 1

Reviewer 1 Report

Comments and Suggestions for Authors

The authors present a manuscript where they describe the preparation of coamorphous drug-drug systems in different molar ratios: felodipine (FEL) or nitrendipine (NTP) using naproxen (NAP) as a combining agent. These solid forms were characterized by other analyses, and dissolution profile, solubility, and shelf stability studies were carried out.

From here, several doubts arise:

1. The individual thermograms shown in Figures 3A and 3B can be provided in the supplementary material since the presence of the Tg is not precisely observed.

2. From Figure 4A and 4B (PXRD), the results indicate that the FEL-NAP coamorphous in all molar ratios do not present any diffraction peak, but in NTP-NAP peaks are observed in the molar ratios (1:1 and 1:2) but in (2:1) there are no reflections. In this case, NTP is in excess. Did you try to obtain an amorphous form of only pure NTP by quench cooling? This is to see if NTP is not reluctant to get amorphous.

3. 3. Concerning Figure 6C, the observed trend of the dissolution profiles was as follows: NTP-NAP (2:1) > amorphous NTP > NTP-NAP (1:2) > NTP-NAP (1:1) > NTP-NAPPM > crystalline NTP. Based on the “spring-parachute” model, because the amorphous NTP presented a better release compared to the solid NTP-NAP forms (1:2 and 1:1). It is known that the “parachute effect,” its prolongation in coamorphous systems, will depend on the nature of the second component. This can delay and prevent the nucleation and crystalline growth of the first component, slowing crystallization. Or can also speed it up. Can you explain why this was observed in NTP-NAP and not in FEL-NAP?

Authors are asked to answer these questions to make the article suitable for publication.

Author Response

  1. The individual thermograms shown in Figures 3A and 3B can be provided in the supplementary material since the presence of the Tg is not precisely observed.

Response: Thanks for your suggestion. The individual thermograms have been provided in the supplementary material.

  1. From Figure 4A and 4B (PXRD), the results indicate that the FEL-NAP coamorphous in all molar ratios do not present any diffraction peak, but in NTP-NAP peaks are observed in the molar ratios (1:1 and 1:2) but in (2:1) there are no reflections. In this case, NTP is in excess. Did you try to obtain an amorphous form of only pure NTP by quench cooling? This is to see if NTP is not reluctant to get amorphous.

Response: Thanks for your question. The figure of PXRD contains an amorphous form of only pure NTP by quench cooling, which do not present any diffraction peak. So, excessive NTP is not a reason of disappearance of its diffraction peak.

  1. Concerning Figure 6C, the observed trend of the dissolution profiles was as follows: NTP-NAP (2:1) > amorphous NTP > NTP-NAP (1:2) > NTP-NAP (1:1) > NTP-NAPPM> crystalline NTP. Based on the “spring-parachute” model, because the amorphous NTP presented a better release compared to the solid NTP-NAP forms (1:2 and 1:1). It is known that the “parachute effect,” its prolongation in coamorphous systems, will depend on the nature of the second component. This can delay and prevent the nucleation and crystalline growth of the first component, slowing crystallization. Or can also speed it up. Can you explain why this was observed in NTP-NAP and not in FEL-NAP?

Response: Thanks for your question. These ratios of FEL-NAP ddCAMs including FEL-NAP (1:1), FEL-NAP (1:2) and FEL-NAP (2:1) systems are completely amorphous. However, for NTP-NAP ddCAMs, only NTP-NAP (2:1) was amorphous. 

Reviewer 2 Report

Comments and Suggestions for Authors

I have reviewed the manuscript pharmaceutics-3296770: Co-amorphization, Dissolution, and Stability of Quench-Cooled Drug-Drug Coamorphous Supersaturating Delivery Systems with RT-Unstable Amorphous Components, which was written by Zhang, Yao and their co-workers.

In this contribution, the authors clearly describe the preparation of drug-drug coamorphous systems (naproxen:felodipine and naproxen:nitrendipine) to improve drugs solubility, dissolution, and stability. They have studied a good topic that may be of interest to Pharmaceutics readers. The methods used for characterization are very well chosen, but the interpretation of the results needs to be improved before the manuscript can be considered for publication.

 I hope the authors find the comments useful in preparing a new version of this manuscript.

1.    The quality of Figure 2 should be improved as the letters (A, B,...) and scale bars are not clearly visible. The morphology of the particles is not clearly visible in Figures 2C, 2D and 2F. Please show an arrow to the location of NAP, FEL and NTP in figure 2D and 2F.

2.    Regarding the following statement: These observations offer valuable insights into the physical and chemical characteristics of quench-cooled systems. What chemical characteristics can be obtained from SEM images?

3.    I think that the section ¨3.4 PXRD analysis¨ should be placed before the sections ¨Micromorphology¨ and ¨Melting Point and Tg Changes¨ to allow a better interpretation of the results.

4.    The structures shown in figures 5C and 5D are not supported by experiments that demonstrate the presence of hydrogen bonds between these groups.

5.    In Line 379-380 change FEL-NAPPM, amorphous FEL, and FEL-NAP quench-cooled systems to: NTP-NAPPM, amorphous NTP, and NTP-NAP quench-cooled systems.

6.    Line 504. The sentence "While FEL-NAP (1:1) exhibited prominent crystal diffraction peaks after 1 month (Fig. 9B1), no such peaks were observed in the FEL-NAP (1:2) (Fig. 9C1) and FEL-NAP (2:1) (Fig. 9D1) systems, suggesting that crystallization took place only in the FEL-NAP (1:1) formulation. " is not supported by the performed experiments. Only amorphous NTP remained amorphous after 1 month. All other samples showed the appearance of some crystalline diffraction peaks.

Author Response

  1. The quality of Figure 2 should be improved as the letters (A, B,...) and scale bars are not clearly visible. The morphology of the particles is not clearly visible in Figures 2C, 2D and 2F. Please show an arrow to the location of NAP, FEL and NTP in figure 2D and 2F.

Response: Thanks for your suggestion. The letters (A, B,...) and scale bars have been improved in SEM images and arrows have been used to point to the location of NAP, FEL, and NTP in Figures 2D and 2F (Figure 4, Line 271).

  1. Regarding the following statement: These observations offer valuable insights into the physical and chemical characteristics of quench-cooled systems. What chemical characteristics can be obtained from SEM images?

Response: Thanks for your suggestion. The statement has been modified to avoid the misunderstanding. Please see the modified statement “These observations offer valuable insights into the physical characteristics of quench-cooled systems” (Line 270).

  1. I think that the section ¨3.4 PXRD analysis¨ should be placed before the sections ¨Micromorphology¨ and ¨Melting Point and Tg Changes¨ to allow a better interpretation of the results.

Response: Thanks for your suggestion. The ¨3.4 PXRD Analysis¨ has been changed to 3.2 PXRD Analysis (Line 220).

  1. The structures shown in figures 5C and 5D are not supported by experiments that demonstrate the presence of hydrogen bonds between these groups.

Response: Thanks for your question. The quench-cooled systems showed the most significant peak shifts, particularly in the FEL-NAP and NTP-NAP systems (Fig. 6C and 6D), strongly suggesting molecular interactions (specifically hydrogen bonding) between the drugs. The possible locations for the formation of hydrogen bonds are illustrated using in Figures 6C and 6D (Line 330).

  1. In Line 379-380 change FEL-NAPPM, amorphous FEL, and FEL-NAP quench-cooled systems to: NTP-NAPPM, amorphous NTP, and NTP-NAP quench-cooled systems.

Response: Thanks for your suggestion. The FEL-NAPPM, amorphous FEL, and FEL-NAP quench-cooled systems have been corrected to NTP-NAPPM, amorphous NTP, and NTP-NAP quench-cooled systems (Line 403).

  1. Line 504. The sentence "While FEL-NAP (1:1) exhibited prominent crystal diffraction peaks after 1 month (Fig. 9B1), no such peaks were observed in the FEL-NAP (1:2) (Fig. 9C1) and FEL-NAP (2:1) (Fig. 9D1) systems, suggesting that crystallization took place only in the FEL-NAP (1:1) formulation. " is not supported by the performed experiments. Only amorphous NTP remained amorphous after 1 month. All other samples showed the appearance of some crystalline diffraction peaks.

Response: Thanks for your question. In this results section, the “no such peaks were observed in the FEL-NAP (1:2) (Fig. 9C1) and FEL-NAP (2:1) (Fig. 9D1) systems” have been corrected to “only a few such peaks were observed in the FEL-NAP (1:2) (Fig. 10C1) and FEL-NAP (2:1) (Fig. 10D1) systems” (Line 534) .

Reviewer 3 Report

Comments and Suggestions for Authors

Authors in the submitted manuscript entitled (Co-Amorphization, Dissolution, and Stability of Quench-Cooled Drug-Drug Coamorphous Supersaturating Delivery Systems with RT-Unstable Amorphous Components) provide a comprehensive study about the preparation of coamorphous form of naproxen with felodipine and nitrendipine. The authors provide a comprehensive study and perform the necessary characterization for prepared formulations. However, there are two issues that need to be resolved before the manuscript is accepted for publication. Firstly, the authors should avoid mentioning all the information in the figures. This will make the manuscript boring for readers. Just mention important findings and the overall conclusion. Moreover, some of the simple principles are illustrated. This manuscript is directed to expertise in areas of interest rather than lecturing to readaers. Secondly, the discussion section is weak and needs to be improved. Please try to compare your findings with those present in the literature.

1.     The introduction is comprehensive and provides sufficient information about the study. However, some sentences need to be rephrased and simplified for the readers.

For example: 

Line 56 -58: To address these challenges, coamorphous (CAM) systems have been as a promising type of SDDS, since CAM systems offer significant

Could be changed to:

Coamorphous (CAM) systems have been developed to address these challenges owing to its advantages over traditional supersaturable ASDs.

Please try to ensure that all sentences are clear and simplified for the readers.

2.     Also, language needs some improvement to address the idea. Some sentences require reading two or three times to be understood.

For example: 

Line 84 – 99: last paragraph usually contains aim of study. These two paragraphs need to be merged and simplified.

3.     Line 103: please move structure to introduction.

4.     Line 106: move table 1 to section 2.2. Usually, composition of formulations is mentioned within the preparation section.

5.     Section 2.7: Did authors mean they use gradient elution method. Please rewrite this section and provide chromatograms in the results section. Please change the title of this section to UPLC method for drug qualification or any relevant title.

6.     Section 2.8. Please mention storage conditions (temperature and humidity).

7.     Section 2.9: please mention if the powder was placed in a capsule or just placed at the beginning of the experiment.

8.     Line 193 – 195: please remove table 2 from the end of the sentence. It is already mentioned at the beginning of the sentence.

9.     Lines 195 – 200: All the data mentioned are present in the table. Please avoid increasing the manuscript length without any demand to avoid boring readers.

10.  Line 242 – 247: please mention this information in the method section.

11.  Line 288 – 292: Please mention why NTP-NAP (2: 1) converted to amorphous compared to NTP-NAP (1:1) and NTP-NAP (1:2)

12.  Figure 6: why did you not perform dissolution for amorphous NTP in 900 mL of 0.25% SDS solution (37 °C)? Also, what is the pH of 0.25% SDS solution? Also, the same symbols for each formation must be maintained constantly in all dissolution profiles to avoid confusion for readers. Please present data as the percentage of drug dissolved rather than concentration.

13.  Please try to summarize the conclusion section. It is very long. Just mention the importance of finding and expected future work.

Author Response

  1. The introduction is comprehensive and provides sufficient information about the study. However, some sentences need to be rephrased and simplified for the readers.

For example: 

Line 56 -58: To address these challenges, coamorphous (CAM) systems have been as a promising type of SDDS, since CAM systems offer significant

Could be changed to:

Coamorphous (CAM) systems have been developed to address these challenges owing to its advantages over traditional supersaturable ASDs.

Please try to ensure that all sentences are clear and simplified for the readers.

Response: Thanks for your suggestion. This section has been simplified and changed (Line 57-58).

  1. Also, language needs some improvement to address the idea. Some sentences require reading two or three times to be understood.

For example: 

Line 84 – 99: last paragraph usually contains aim of study. These two paragraphs need to be merged and simplified.

Response: Thanks for your suggestion. This section has been to be merged and simplified (Line 84-97) .

  1. Line 103: please move structure to introduction.

 Response: Thanks for your suggestion. This section has been moved (Line 98).

  1. Line 106: move table 1 to section 2.2. Usually, composition of formulations is mentioned within the preparation section.

  Response: Thanks for your suggestion. This section has been moved (Line 119).

  1. Section 2.7: Did authors mean they use gradient elution method. Please rewrite this section and provide chromatograms in the results section. Please change the title of this section to UPLC method for drug qualification or any relevant title.

  Response: Thanks for your suggestion. This section used isocratic elution method. The chromatograms have been provided in the results section. The title of this section has been changed to “HPLC method for drug qualification” (Line152).

  1. Section 2.8. Please mention storage conditions (temperature and humidity).

  Response: Thanks for your suggestion. This section has been added (Line165).

  1. Section 2.9: please mention if the powder was placed in a capsule or just placed at the beginning of the experiment.

 Response: Thanks for your question. The powder was placed at the beginning of the experiment (Line173).

  1. Line 193 – 195: please remove table 2 from the end of the sentence. It is already mentioned at the beginning of the sentence.

 Response: Thanks for your suggestion. Table 2 has been removed from the end of the sentence (Line 201).

  1. Lines 195 – 200: All the data mentioned are present in the table. Please avoid increasing the manuscript length without any demand to avoid boring readers.

 Response: Thanks for your suggestion. This section has removed all the data in the manuscript (Line 200-211).

  1. Line 242 – 247: please mention this information in the method section.

 Response: Thanks for your suggestion. Section of 2.5 has been added (Line 138-141).

  1. Line 288 – 292: Please mention why NTP-NAP (2: 1) converted to amorphous compared to NTP-NAP (1:1) and NTP-NAP (1:2)

Response: Thanks for your suggestion. This section has been changed (Line 241).

  1. Figure 6: why did you not perform dissolution for amorphous NTP in 900 mL of 0.25% SDS solution (37 °C)? Also, what is the pH of 0.25% SDS solution? Also, the same symbols for each formation must be maintained constantly in all dissolution profiles to avoid confusion for readers. Please present data as the percentage of drug dissolved rather than concentration.

 Response: Thanks for your question. 1) I think you ask to “why did you not perform dissolution for amorphous NAP in 900 mL of 0.25% SDS solution (37 °C)?” rather than amorphous NTP. Because amorphous NAP is unstable under RT conditions, dissolution experiments of amorphous NAP at 37 °C cannot be conducted. 2) The pH of 0.25% SDS solution is 7.2; 3) The data as the percentage of drug dissolved has been presented.

  1. Please try to summarize the conclusion section. It is very long. Just mention the importance of finding and expected future work.

   Response: Thanks for your question. We have re-summarized this conclusion section clearly and simply (Line 651-670).

Round 2

Reviewer 1 Report

Comments and Suggestions for Authors

I consider that the authors attended to the observations made in the first manuscript sent and that the current form of the document is suitable for publication in the journal Pharmaceutics.

Author Response

I consider that the authors attended to the observations made in the first manuscript sent and that the current form of the document is suitable for publication in the journal Pharmaceutics.

Response: Thanks for your recognition. 

Reviewer 2 Report

Comments and Suggestions for Authors
  1. Response: Thanks for your question. The quench-cooled systems showed the most significant peak shifts, particularly in the FEL-NAP and NTP-NAP systems (Fig. 6C and 6D), strongly suggesting molecular interactions (specifically hydrogen bonding) between the drugs. The possible locations for the formation of hydrogen bonds are illustrated using in Figures 6C and 6D (Line 330).

Suggestions for Authors: Given that experiments do not support the structures, it is necessary to clarify in the figure and in the manuscript that it is a potential location for hydrogen bonds.

Author Response

  1. Response: Thanks for your question. The quench-cooled systems showed the most significant peak shifts, particularly in the FEL-NAP and NTP-NAP systems (Fig. 6C and 6D), strongly suggesting molecular interactions (specifically hydrogen bonding) between the drugs. The possible locations for the formation of hydrogen bonds are illustrated using in Figures 6C and 6D (Line 330).

Suggestions for Authors: Given that experiments do not support the structures, it is necessary to clarify in the figure and in the manuscript that it is a potential location for hydrogen bonds.

Response: Thanks for your suggestion. A potential location for hydrogen bonds have been clarified in the Figures 6C/D and in the manuscript (Line 330 and Line 350).

Reviewer 3 Report

Comments and Suggestions for Authors

The authors improved the revised version of the manuscript. However, there are minor issues needs to be improved before acceptance for publication.

1.     Section 2.7. Please summarize all the necessary information in the table. Kindly present the mobile phase, calibration range, and detection wavelength for each drug. This will improve the presentation of your data.

2.     Section 2.8. It is relative humidity, not humidity. Also, please provide a unit for relative humidity (%).

3.     Section 2.9: Mass equivalents (50 mg).

This quantity refers to the prepared formulation or drug.

If this quantity refers to a drug (NAP), please mention it.

If for the formulation, this mean inequivalent amount of drug was used because you prepared different ratios between NAP and (FEL and NTP). Even if you use this approach, please mention it and consider it in future work.

Author Response

The authors improved the revised version of the manuscript. However, there are minor issues needs to be improved before acceptance for publication.

  1. Section 2.7. Please summarize all the necessary information in the table. Kindly present the mobile phase, calibration range, and detection wavelength for each drug. This will improve the presentation of your data.

Response: Thanks for your suggestion. The mobile phase, calibration range, and detection wavelength for each drug have been presented in table 2(Line 158-161 ).

  1. Section 2.8. It is relative humidity, not humidity. Also, please provide a unit for relative humidity (%).

 Response: Thank you for your reminding.The relative humidity have been corrected and the unit for relative humidity have been provided(Line 164).

  1. Section 2.9: Mass equivalents (50 mg).

This quantity refers to the prepared formulation or drug.

If this quantity refers to a drug (NAP), please mention it.

 Response: Thanks for your suggestion. This quantity refers to the prepared formulation or drug(Line 170-173).

4.If for the formulation, this mean inequivalent amount of drug was used because you prepared different ratios between NAP and (FEL and NTP). Even if you use this approach, please mention it and consider it in future work.

 Response: Thanks for your valuable suggestion. We have mentioned it  and considered it in future work. Please see "On the other hand, by preparing different representative ratios between NAP and (FEL and NTP) for various studies, theoretical guidance and data support have been provided for further clinical combination therapy based on  these drug-drug pairs. The actual dosage of drug-drug in clinical practice still  needs to be determined and regulated according to  patients’ condition, and this will be another focus in our future work.”  (Line 670-675)
